# Association between care burden, depression and personality traits in Alzheimer's caregiver: A pilot study

Anna Vespa[1]*, Roberta Spatuzzi[2], Paolo Fabbietti[3], Martina Penna[1], Maria Velia Giulietti[1]

1 Department of Neurology, IRCCS INRCA -National Institute of Science and Health for Aging, Ancona, Italy, 2 Department of Mental Health, ASP-Healthcare Service, Basilicata, Potenza, Italy, 3 Bio-Statistical Center, IRCCS INRCA-National Institute of Science and Health for Aging, Ancona, Italy

* a.vespa@inrca.it

## Abstract

### Introduction

In this study correlations between care burden, depression, and personality at intrapsychic level in caregivers of Alzheimer's disease patients were evaluated.

### Materials and methods

Caregivers: n.40. Tests: Social-schedule; CBI; BDI; SASB-Structural-Analysis of Social Behaviours- Form-A- intrapsychic behaviours (8 Cluster); ECOG. Patients:MMSE. Statistical analysis: Chi-squared test; Anova one way F test; Pearson's R coefficient.

### Results

Correlations: CBI-total and NPI(p < .001); CBI-total—ECOG (p = .042); CBI-total—BDI(p< .001); CBI- total-SASB-Cl7(p = .014); SASB-CL8(p<0.000); BDI and SASB-Cl 2 (p = .018), SASB-Cl 3 (p = .004), SASB-Cl7(p < .000), SASB-CL8 (p < .000). High CBI is correlated with high depression, neuropsychiatry symptoms, low cognitive patient's functions. Caregivers have the following intrapsychic behaviors: poor self-care, poor ability to take care of themselves; they exercise control over themselves and do not consider and/or ignore their basic needs at emotional and physical levels. These intrapsychic behaviours are indicators of depression (SASB Model) and are correlated with high care burden–CBI and high depression-CDQ.

### Discussion

Care burden is closely related to the depression and individual personality (intrapsychic experience) of the caregiver. This may reveal a source of strength and may suggest areas of multidimensional and psychotherapeutic interventions.

**Data Availability Statement:** The data used in this study were collected by the authors. The authors are employees of the INRCA, thus the data belong to both the authors and the INRCA. Requests for

data can be sent to Scientific Direction INRCA, National Institute of Science and Health for Aging, Ancona Italy, email:direzionescientifica@inrca.it.

**Funding:** The author(s) received no funding for this work.

## Introduction

Alzheimer's disease is the most common dementia in the elderly, resulting in a progressive decline of cognitive functions, such as attention, memory, language, reasoning. The patient presents an impairment of spatial-temporal orientation and significant relationships, with the failure to recognize loved ones. A progressive disease such as Alzheimer's disease leads to a profound change in the lifestyle of the entire family system involving the emotional sphere, different problems in various stages, practical and organizational problems [1–5].

All these aspects can trigger new conflicts due to fatigue, economic problems or decisions to be made. They can also reactivate old tensions that sometimes lead to definitive relationship breakdowns [5–7]. Caring for a person with dementia is a very demanding task both emotionally and physically: it can limit emotional resources and the caregiver can exhibit anxious-depressive symptoms with somatic problems that often limit the ability to care [1–5]. Attention to the caregiver therefore becomes increasingly important.

Behavioral and psychological symptoms of dementia (BPSD) affect most patients during disease progression [8–10] and involve the patient, caregiver, and their environment. BPSDs are more stressful for caregivers [11, 12], also if they differ in their emotional responses to them. Thus subjective factors and individual differences between caregivers in care experience are important [11–17] and may play a role in determining the caregiver's perception of the patient's problems and the consequent adequate or inadequate response. In fact, the individual characteristics of personality are closely related to the way of reacting to the condition of caregiving and to the management of the individual's resources in the face of problematic life events.

The caregiver can experience emotions ranging from anger at the new condition of their loved one, which is difficult to accept, to anxiety, distress, insomnia, feelings of helplessness. All these conditions lead to a worsening of the quality of life [15]. Extensive literature suggests that chronic stress, associated with caring for a loved one with dementia, can also negatively impact the physical health of the caregiver, with an increased risk of developing diseases [14]. As for mood, these caregivers experience depressive symptoms and emotions of fear, hostility, and sadness [11–14].

Caregivers also report low access to psychological resources, such as social support, personal skills, enjoyable activities, and perceived self-efficacy in solving problems. Living with an Alzheimer's patient leads to a marked and significant restriction of activities [12]. So the caregiver' s state of health and well-being is directly connected with the care burden which influence the quality of life and the mood, with particular reference to depressive symptoms [14]. Moreover caregivers are at greater risk of compromising both physical health and subjective psychosocial well-being [15].

In this context, the individual characteristics of the personality are closely linked to how an individual reacts to the management of resources in the face of problematic life events.

The way of coping with stressful situations such as, care burden, anxiety and depressed mood [2, 3, 17] are both partly determined by the personality of the caregiver [11, 15, 18, 19] and can result in a positive or negative adaptation. It is essential that the physical and emotional stress is recognized early on by both the clinician and the family unit so that a timely bio-psycho-social intervention can be planned, aimed at addressing the personal difficulties of the caregiver [16–18, 20, 21]. Based on these considerations, studying the personality and the relationship with the stressors that affect these caregivers may make it possible to plan and provide the support and care they need.

To our knowledge, no studies have analyzed the personality of caregivers of Alzheimer' patients and no prior study exists on the intrapsychic personality traits of caregivers, using the

SASB Model of L.S. Benjamin. We believe that the Person-centered approach, with clusters analysis, which conceptualizes the personality as "a related system of different traits" allows the identification of latent classes (subgroups) of individuals with distinct personality size profiles, and is the most appropriate in this context. For this reason, in this study we have implemented the SASB circumplex model used to describe personality from normal to pathological and defines interpersonal and intrapsychic behaviors by three underlying dimensions: focus (other, self, introject), affiliation-hostility (love-hate), and interdependence-independence (enmeshment-differentiation). SASB applications extend from research to practice: assessment, treatment of intrapsychic problems and verification of psychotherapeutic intervention [22, 23].

In particular, we tried to identify psychological vulnerabilities (intrapsychic behaviours), and we investigated the role that care burden plays in relation to depression and in which condition (neuropsychiatry symptoms). This may reveal a source of strength and may suggest areas of therapeutic intervention.

## Materials and methods

Forty primary family caregivers of patients with diagnosed Alzheimer's disease, volunteered to participate in the study protocol. The caregivers were selected from the Italian Hospital-IRCC-S-INRCA-National-Institute of Science, from September 2019 to August 2020 (three-month-break during the Corona-Virus lock-down). The protocol was approved by Bioethical Advisory Committee of IRCCS-INRCA, Ancona, Italy: code no. 18020. Eligibility was based on the following criteria: having a relative with diagnosis of Alzheimer disease; being at least 18 years of age; having no health problems, diagnosis of cancer and/or no neurological or cognitive impairments; being identified as the main caregiver, either by the patient or self-identified; being proficient in the Italian language; providing a written, informed consent. Seventy seven caregivers, meeting all inclusion criteria, were consecutively recruited at the Neurology Departments of INRCA-IRCCS National Institute of Science and Health in Ancona, Italy and asked to participate in the study. After completing this initial medical examination they were referred to the investigator. Only fifty two caregivers decided to participate and signed a consensus form, after detailed explanation by the physicians at the clinics. The caregivers were interviewed by two psychotherapists, working in the hospital, with one year training in learning and administering the SASB Model, in particular the Form-A. The caregivers filled out the questionnaires (which included socio-demographic and clinical variables) during the interview. Twelve caregivers did not answer all the questions in the questionnaires: it was therefore decided not to consider them for the analysis. Consequently our sample is of forty caregivers.

All caregivers were asked to complete the following tests:

**Social schedule**: data on sex, age, marital status, educational level, profession.

**SASB–Form-A**-Questionnaire by L.S. Benjamin [22, 23] **Structural Analysis of Social-Behaviour** that describes the psychic processes of the personality structure at the intrapsychic level (S1 Appendix) by three underlying dimensions: focus (other, self, introject), affiliation-hostility (love-hate), and interdependence-independence (enmeshment-differentiation). This test is short (36 questions) and has the appropriate reliability and validity to evaluate intrapsychic behaviours and is validated on the basis of DSMIV. The Italian version has been validated on the Italian population. The SASB Form-A -Questionnaire describes the structure of personality from normal to pathological: is rated on a 10-point scale (0 = never to 10 = always). The 36 questions of Form-A are grouped by a specific score correction into the following 8 clusters of intrapsychic- behaviours (Oneself):

SASB- Cl1 = Autonomy-Assertive and separating.

SASB-Cl2 = Autonomy and love-Self-accepting and exploring.

SASB-Cl3 = Love -Self-supporting and appreciative.

SASB-Cl4 = Love and control-Self-care and development.

SASB-Cl5 = Control-Self-regulating and controlling.

SASB- Cl6 = Control and hate-Self-critical and oppressive.

SASB-Cl7 = Hate -Self-refusing and annulling.

SASB-Cluster (Cl) 8 = Hate and autonomy-Self-negligent and mentally absent.

**The Everyday Cognition (ECOG) is** a subjective scale that assesses cognitive function and IADL related to 6 domains [24]. The ECOG consists of 39 items, 17 of which are specific to IADL (this was determined by an experienced clinician, GAM). It is administered separately to the participant and to the informant. The score range for each item is 1–4 (higher scores indicate greater impairment; 1 = better or no change compared to 10 years ago; 2 = question-able/occasionally worse; 3 = consistently a little worse; and 4 = consistently much worse). The total score on the ECOG has been previously shown to successfully discriminate between CN elderly and individuals with MCI.

**The Caregiver Burden Inventory (CBI)** [25] is a 24 items tool for assessing the care bur-den. It analyzes the multidimensional, elaborate aspects of the caregiver' s experience of patients with Alzheimer's disease and related dementias.

The CBI is a self-report tool, compiled by the principal caregiver (i.e. family member or operator who most bears the burden of care). Divided into 5 sections, it allows to evaluate the following different stress factors: 1) Objective burden (items 1–5); Time dependent: the bur-den in relation to the time that the caregiver has to devote to his family member; 2) Develop-mental burden: the caregiver's perception of feeling cut off from expectations and opportunities of their peers (items 6–10); 3) Physical burden: the feelings of chronic fatigue and somatic health problem (items 11–14); 4) Social burden: the perception of a role conflict (items 15–19); 5) Emotional burden: the feelings towards the patient, that can be induced by unpredictable and bizarre behavior (items 20–24).

There are 5 items for each dimension, except for the physical burden, which has four items.

Therefore the total score for Time-dependence, Developmental, Social and Emotional bur-den ranges from 0 to 20, except for physical burden where scores range from 0 to 16. The score for physical burden could be multiplied by 1.25 to obtain an equivalent score out of 20. The total score range ranges from 0 to 96: a score> 36 indicates a risk of "exhaustion" while scores close to or slightly above 24 indicates the need to seek some form of respite care.

## Beck depression inventory, 2nd version (BDI-II)

The Beck Depression Inventory (BDI) by Aaron T. Beck [26, 27], is a 21-question multiple-choice self-report inventory and one of the most widely used psychometric tests for measuring the severity of depression on a 4-point scale ranging from 0 to 3. The BDI-II is scored by sum-ming the highest ratings for each of the 21 items. Thus, the total score can range form 0 to 63.

The internal consistency of the BDI-II was demonstrated to be good (Cronbach's $\alpha$ = .91) and the 1-week test-retest reliability was shown to be high.

**The Mini-Mental State Examination** (MMSE) [28, 29] is the most commonly used demen-tia screening tool. The MMSE consists of 30 questions and includes tests of orientation, con-centration, attention, verbal memory, naming and visuospatial skills. Scores for items are summed together into a total score. Higher scores indicate better cognition and the maximum

score is 30. The total score, given by the sum of the scores that the patient has obtained for each item, can range from a minimum of 0 (maximum cognitive impairment) to a maximum of 30 (no cognitive impairment). The cutoff score is 23–24, and most non-demented seniors rarely score below 24.

## Patients cognitive evaluation and diagnosis of patients with Alzheimer disease

The diagnosis of Alzheimer's disease was confirmed based on the criteria of the Diagnostic Statistical Manual of Mental Disorders (DSM-IV) [30] and of the National Institute of Neurological and Communicative Disorders and Stroke and the Alzheimer's Disease and Related Disorders Association (NINCDS-ADRDA) [31].

**H. The Neuropsychiatric-Inventory–Questionnaire** (NPI-Q) [32, 33] provide a brief assessment of neuropsychiatric symptomatology. The NPI-Q is designed to be a self-administered questionnaire completed by informants about patients for whom they care. Each of the 12 NPI-Q domains contains a survey question that reflects cardinal symptoms of that domain. Responses to each domain question are "Yes"(present) or "No" (absent). If the response to the domain question is "No", the informant goes to the next question. If "Yes", the informant then rates both the Severity of the symptoms present within the last month on a 3-point scale and the associated impact of the symptom manifestations on them (i.e. Caregiver Distress) using a 5-point scale. The NPI-Q provides symptom Severity and Distress ratings for each symptom reported, and total Severity and Distress scores reflecting the sum of individual domain scores.

## Statistical analysis

Data were reported by mean ± standard deviation for continuous variables and by number and percentage for categorical ones. Chi-squared test was used to compare categorical variables, while Anova one way F test, or independent samples's t-test when relation between categorical and continuous variables was studied. Pearson's R coefficient was used to compare two continuous variables. The probability p-value <0.05 was considered statistically significant. Statistical analysis was carried out using SPSS for Win V24.0 statistical software package (SPSS Inc, Chicago, IL, USA).

## Results

The following parameters were collected about caregivers: gender, marital status, educational level. relationship with the patient, length of time care(months), and time of daily care(hours). No differences emerged for the burden by gender and age, civil status (Table 1). The demographic characteristics of the Alzheimer's patients were described in Table 2.

The patient's cognitive status was screened by means of the Mini-Mental-State-Examination (MMSE).

Alzheimer's disease patients of our sample have moderate (85%) to severe (15%) cognitive impairment (MMSE). No correlations emerged between MMSE and CBI.

## Correlations: CBI 8 (Burden) and NPI (Neuropsychiatry Inventory)

Total-CBI(CBI-T) is correlated with total NPI(p < .001); with delirium (p = .027); depression/ dysphoria(p = .039); anxiety(p < .000); apathy / indifference(p = .027); aberrant motor activity (p < .000); sleep disorders (p = .002); disorders of appetite and nutrition(p = .038) (Table 3). There was no correlation between CBI and hallucinations, agitation, euphoria / exaltation, disinhibition and irritability / lability (Table 3).

**Table 1. Socio-demographic characteristics of the caregivers.**

| Caregivers | |
|---|---|
| **Gender, n(%)** | |
| Female | 25 (62.5%) |
| Male | 15 (37.5%) |
| **Age, mean±sd** | 59.3±9.1 |
| Female | 57.76±7.49 |
| Male | 61.87±11.10 |
| **Age classes, n(%)** | |
| 46–60 | 24 (60.0%) |
| 61–70 | 12 (30.0%) |
| 71–87 | 4 (10.0%) |
| **Marital status, n(%)** | |
| Not -married | 3 (7.5%) |
| Married | 32 (80.0%) |
| Widowed | 1 (2.5%) |
| Separated / divorced | 4 (10.0%) |
| **Educational Level, n(%)** | |
| Primary School | 2 (5.0%) |
| Secondary School | 10 (25.0%) |
| High School | 22 (55.0%) |
| University | 6 (15.0%) |
| **Degree of Kinship, n(%)** | |
| Wife | 4 (10.0%) |
| Husband | 2 (5.0%) |
| Son /daughter | 32 (80.0%) |
| Other | 2 (5.0%) |

The following correlations emerged between burden (CBI) and degree of functional deterioration (ECOG): CBI-T and ECOG 4(p = 0.042); T/Dep-B and ECOG-4 (p = 0.016) (Table 4).

## Correlations: CBI and educational level

The caregiver with primary school showed more DEv-B(p = .042) than the others while the caregivers with university degree had more emotional burden (Em-B) (p = .043) (Table 5).

## Correlations: Care burden and time of care

Caregivers whose care time is continuous showed a correlation with CBI / Dep- B (p = .003); the care time of 3 hours / day is correlated with CBI Tot (p = .003); CBI / Dep / B (p <000); CBI/ DEv-B (p = .002); the time of care time of 7–12 hours / day hours / day is correlated with CBI- T (p = .012); CBI Dev-B (p = .050- tendency); CBI- T / Psy-B(p < .005); CBI EMO-B (p = .043) (Table 6).

## Correlations: Care burden and degree of kinship

Spouses had a greater care burden than other caregivers (son/daughter or other carer) (p = .013), CBI/DEp (p = .017), CBI/Dev-B(p < .001), CBI/ Phys-B(p = .011) (Table 7).

## Correlations: Care burden (CBI), depression (BDI)

Caregivers with greater care burden(CBI) had higher levels of depression(BDI) (p< .001) (Table 8).

**Table 2. Socio-demographic characteristics of the Alzheimer's patients (gender and age).**

| Patients | |
|---|---|
| **Gender, n(%)** | |
| Female | 30 (75.0%) |
| Male | 10 (25.0%) |
| **Age, mean±sd** | 83.5±6.3 |
| **Age classes, n(%)** | |
| 64–74 | 3 (7.5%) |
| 75–85 | 20 (50.0%) |
| 86–94 | 17 (42.5%) |
| **Frequency of Neuropsychiatric Symptoms** | |
| **Delirium, n(%)** | 24 (60.0%) |
| **Hallucinations, n(%)** | 20 (50.0%) |
| **Agitation, n(%)** | 29 (72.5%) |
| **Depression / dysphoria, n(%)** | 32 (80.0%) |
| **Anxiety, n(%)** | 24 (60.0%) |
| **Euphoria / elation, n(%)** | 8 (20.0%) |
| **Apathy / indifference, n(%)** | 33 (82.5%) |
| **Disinhibition, n(%)** | 17 (42.5%) |
| **Irritability / lability, n(%)** | 28 (70.0%) |
| **Aberrant motor activity, n(%)** | 11 (27.5%) |
| **Sleep disorders, n(%)** | 23 (57.5%) |
| **Disorders of appetite and nutrition, n(%)** | 19 (47.5%) |

## Correlations: Care burden (CBI), and SASB Cls

The following correlations between the SASB clusters and CBI emerged: SASB-Cl3 (p = .058—tendency), SASB-Cl7 (p = .014) and SASB-CL8 (p < .000) (Table 9).

Caregivers showed the following intrapsychic behaviors correlated with burden: poor self-care (Cl3) (tendency), self-criticism, neglect and / or not consideration of their basic emotional and physical needs (Cl7, Cl8).

**Table 3. Correlation between caregiver's care burden (CBI) and patient's neuropsychiatry symptoms (NPI).**

| | CBI, mean±sd | p | Pearson's r |
|---|---|---|---|
| **Total NPI** | 30.35±22.84 | .000 | .552** |
| **Delirium** | 3.15±3.86 | .027 | .349* |
| **Hallucinations** | 2.30±3.42 | .222 | .197 |
| **Agitation** | 3.45±3.59 | .108 | .258 |
| **Depression / dysphoria** | 4.32±3.25 | .039 | .328* |
| **Anxiety** | 3.42±3.52 | .000 | .536** |
| **Euphoria / elation** | 0.40±1.08 | .574 | .092 |
| **Apathy / indifference** | 3.15±3.86 | .027 | .349* |
| **Disinhibition** | 1.40±2.58 | .107 | .258 |
| **Irritability / lability** | 2.45±2.52 | .243 | .189 |
| **Aberrant motor activity** | 1.42±2.64 | .000 | .572** |
| **Sleep disorders** | 3.00±3.60 | .002 | .480** |
| **Disorders of appetite and nutrition** | 1.87±2.61 | .038 | .330* |

**Table 4. Correlation between burden (CBI) and degree of functional deterioration (ECOG).**

|   | T/Dep-B | Dev-B | Phys-B | Soc-B | Emo-B | CBI Tot. |
|---|---------|-------|--------|-------|-------|----------|
| 0 | - | - | - | - | - | - |
| 1 | 5.00 | 1.00 | 2.50 | .00 | .00 | 8.50 |
| 2 | 7.00±5.56 | 5.85±5.58 | 4.82±4.29 | 2.85±2.73 | 3.00±2.88 | 23.53±16.48 |
| 3 | 11.36±4.66 | 7.13±4.83 | 5.90±4.10 | 2.27±3.02 | 1.40±1.43 | 28.09±11.05 |
| 4 | 14.60±4.74 | 11.00±6.30 | 7.62±6.13 | 6.30±6.29 | 2.80±4.13 | 42.32±23.80 |
| p | .016 | .112 | .545 | .072 | .317 | .042 |

## Correlations: Depression (BDI) and SASB Cls

The following correlations emerged between the SASB clusters and BDI: SASB–Cl 2 (p = .018), SASB-Cl3 (p = .004), SASB-Cl7(p < .000), SASB-CL8 (p < .000) (Table 9).

Caregivers showed the following intrapsychic behaviors correlated with depression: poor self-acceptance (Cl2), poor self-care (Cl3), self-criticism, neglect and / or not consideration of their basic emotional and physical needs (Cl7, Cl8). (Table 9).

These intrapsychic behaviours in SASB model are indicators of depression (SASB: medium low SASB Cl 2, low -Cl 3, high Cl7, high Cl8).

## Discussion

From our results control variables such as age, and sex, were not associated with care burden CBI but we hypothesize that our results could be due to the small sample: gender and age could have different influences on giving assistance. As for the degree of kinship, spouses have more CBI-T. Furthermore, caregivers with primary school education showed more DEV-burden while undergraduate caregivers have more emotional-burden than the other categories.

CBI is correlated with total NPI and, in particular, with the following dimensions: delirium; depression/dysphoria; anxiety, apathy / indifference; aberrant motor activity; sleep disorders; disorders of appetite and nutrition. Our results are in agreement with those of Liu [8] who affirms that depression, anxiety and sleep problems are the main challenges that are faced by family caregivers of patients with Alzheimer's disease.

Caregiver with primary school education showed more Developmental Burden than the others while caregivers with a university degree had more Emotional burden. The differences in educational level that emerged from our study are not supported by other studies. The results of Delfino [34] showed that female caregivers have a significant difference for burden but no differences emerged for the educational level. Further studies may clarify this topic.

**Table 5. CBI (Burden) and educational level.**

|   | CBI-Educational Level | | | | | | | | | | | |
|---|---|---|---|---|---|---|---|---|---|---|---|---|
|   | Primary School vs Other | | | Secondary School vs Other | | | High School vs Other | | | University vs Other | | |
|   | Categories | | | Categories | | | Categories | | | Categories | | |
|   | mean±sd | mean±sd | p | mean±sd | mean±sd | p | mean±sd | mean±sd | p | mean±sd | mean±sd | p |
| CBI tot | 46.87±1.94 | 29.49±0 | .169 | 31.37±15.58 | 30.02±18.04 | .834 | 28.6±14.55 | 32.51±20.35 | .483 | 29.62±29.59 | 30.49±14.83 | .911 |
| T/Dep-B | 17.5±0.7 | 10.92±5.3 | .091 | 11.5±3.24 | 11.16±5.95 | .868 | 10.9±5.91 | 11.66±4.75 | .663 | 10±6.38 | 11.47±5.24 | .543 |
| Dev-B | 10.92±5.3 | 7.31±5.43 | .042 | 7.7±5.33 | 7.73±5.76 | .987 | 7.54±5.13 | 7.94±6.25 | .826 | 5.83±7.3 | 8.05±5.3 | .376 |
| Phys-B | 11.87±0.88 | 5.75±4.59 | .071 | 7.37±5.41 | 7.37±5.41 | .311 | 5.05±3.23 | 7.29±5.85 | .134 | 5.62±7.19 | 6.13±4.23 | .807 |
| Soc-B | 1±0 | 3.44±4.37 | .439 | 3.6±4.4 | 3.23±4.32 | .819 | 3.18±3.88 | 3.5±4.85 | .819 | 4.16±6.49 | 3.17±3.9 | .609 |
| Emo-B | 1±1.41 | 2.05±2.69 | .589 | 1.2±1.03 | 2.26±2.95 | .274 | 1.9±2.28 | 2.11±3.08 | .813 | 4±4.85 | 1.64±1.95 | .043 |

**Table 6. CBI- time of care (hours of assistance).**

| | CBI-ore di assistenza | | | | | | | | | | | |
| --- | --- | --- | --- | --- | --- | --- | --- | --- | --- | --- | --- | --- |
| | Continuous care- 24h vs Other Categories | | | Day care 3h vs Other Categories | | | Day care 4-6h vs Other Categories | | | Day care 7-12h vs Other Categories | | |
| | mean±sd | mean±sd | p | mean±sd | mean±sd | p | mean±sd | mean±sd | p | mean±sd | mean±sd | p |
| CBI tot | 37.37±12.93 | 28.02±0 | .14 | 22.14±13.67 | 37.79±17.08 | .003 | 28.4±11.13 | 30.64±18.09 | .790 | 46.33±24.3 | 27.54±14.45 | .012 |
| T/Dep-B | 15.4±3.3 | 9.86±5.23 | .003 | 7.73±4.31 | 14.42±4.13 | 0 | 12.4±4.21 | 11.08±5.54 | .615 | 14.5±5.31 | 10.67±5.24 | .109 |
| Dev-B | 10.5±5.83 | 6.8±5.28 | .069 | 5±4.44 | 10.19±5.45 | .002 | 7.6±2.96 | 7.74±5.9 | .958 | 11.83±6.36 | 7±5.21 | .050 |
| Phys-B | 6.87±3.78 | 5.79±4.96 | .533 | 4.93±4.47 | 7.08±4.71 | .149 | 3±2.59 | 6.5±4.76 | .118 | 10.83±4.91 | 5.22±4.15 | .005 |
| Soc-B | 3.1±3.92 | 3.4±4.46 | .851 | 2.52±3.65 | 4.04±4.76 | .269 | 4.6±5.02 | 3.14±4.22 | .485 | 5.16±6.27 | 3±3.88 | .260 |
| Emo-B | 1.5±1.17 | 2.17±2.97 | .496 | 1.94±2.61 | 2.04±2.72 | .906 | 0.8±0.44 | 2.17±2.78 | .283 | 4±4.51 | 1.64±2.07 | .043 |

Caregivers who provide 7–12 hours of care have more CBI-Tot, CBI Developmental, Physical, Social and Emotional CBI EMO-B Burden than caregivers with less hours of assistance. Pudelewicz [35] affirms that there was a significant correlation between the feeling of burden and the caregiver's lack of free time, the number of hours devoted to day care.

As for the degree of kinship, spouses have more CBI-T. Probably it may be due to the greater care time they provide to the patient, reduction or lack of social relations and leisure time activities. This result, however, must be taken with caution as our sample is small and unbalanced: the spouse constitutes the majority in the civil status category (80%). For some author it is the severity of the disease that plays an important role in the reorganization of the family environment in families caring for not institutionalized patients with Alzheimer, and weight on quality of life and burden, regardless of the type of relationship [36]. Moreover all caregivers have to face the degeneration of the patient's personality and live with a person who is no longer fully the loved one and this implies a trauma: we think it might weigh on the stress of all caregivers, especially spouses. Further studies may address these issues more deeply.

Caregivers with greater care burden have higher levels of depression.

Moreover the objective of this study was to determine the association between care burden (CBI), depression (BDI) and intrapsychic and interpersonal behaviors of caregivers of patients with Alzheimer's disease to better understand the caregiving conditions. The correlations which emerged suggest the following considerations.

Caregivers who are engaged in the following intrapsychic behaviors: low self-esteem and low self-acceptation (SASB Cl3), poor ability to take care of themselves and re-consolidate themselves (Cl3) not considering and / or ignoring and neglecting their basic needs at emotional and physical levels (Cl7,Cl8) manifest high depression. Our results showed that these caregivers are more likely to experience high CBI.

**Table 7. CBI (Burden) and degree of kinship.**

| | Spouse vs Other Categories | | | Son/Daughter vs Other Categories | | |
| --- | --- | --- | --- | --- | --- | --- |
| | mean±sd | mean±sd | p | mean±sd | mean±sd | p |
| CBI tot | 46.25±8.49 | 27.55±0 | .013 | 28.79±16.71 | 36.62±19.22 | .257 |
| T/Dep-B | 16±1.78 | 10.41±5.36 | .017 | 10.71±5.34 | 13.37±5.26 | .215 |
| Dev-B | 14.16±4.35 | 6.58±5.02 | .001 | 6.96±4.92 | 10.75±7.32 | .087 |
| Phys-B | 10.41±2.45 | 5.29±4.56 | .011 | 5.54±4.57 | 8.12±4.77 | .166 |
| Soc-B | 3.66±5.12 | 3.26±4.21 | .836 | 3.46±4.26 | 2.75±4.65 | .678 |
| Emo-B | 2±1.26 | 2±2.82 | 1.000 | 2.09±2.88 | 1.62±1.3 | .659 |

**Table 8. Correlation between caregiver's care burden (CBI) and depression (BDI-II).**

|  | CBI Tot mean±sd | p | F |
|---|---|---|---|
|  |  | < .001 | 12.285 |
| Absent (0–13) | 24.11±13.66 |  |  |
| Mild (14–19) | 38.65±10.00 |  |  |
| Moderate (20–29) | 47.08±2.46 |  |  |
| Severe (30–63) | 88.75±0.0 |  |  |

Furthermore, CBI is significantly correlated with depression in these caregivers. Our findings support previous researches [8] suggesting that caregivers who score high CBI have also high depression (BDI) while those with low CBI have also a better mood.

If depression is a consequence of the care burden, as emerges from this study, the accentuation of intrapsychic behaviors could also be a consequence since the intrapsychic modalities described are an indication of depression.

It is important to understand whether these intrapsychic behaviors pre-exist the condition of caregiving or are a consequence of it: the two hypotheses can both be true as the care cost implies the tendency to fold in on oneself and to become depressed, and vice versa the subjective personality can have an influence on the way of reacting to the condition of caregiving and on the mood, in a circle of mutual influence.

Further studies may examine these issues.

Our study represents a first look at the relationships between the intrapsychic aspects of the personality structure of caregivers and the care burden and depression.

In particular the dimensions of not taking care of themselves and their basic needs at emotional and physical levels, are associated with high BDI and CBI.

The continuity of research on the relation between CBI and depression will enable to better understand the effect of perceived support toward physical and mental health, along with a

**Table 9. Correlations between caregiver's burden (CBI).**

|  |  | CBI Tot | BDI Tot |
|---|---|---|---|
| SASB Cl 1 | Pearson Correlation | .063 | -.042 |
|  | Sig. (2-tailed) | .702 | .798 |
| SASB Cl 2 | Pearson Correlation | -.251 | -.374* |
|  | Sig. (2-tailed) | .119 | .018 |
| SASB Cl 3 | Pearson Correlation | -.302 | -.449** |
|  | Sig. (2-tailed) | .058 | .004 |
| SASB Cl 4 | Pearson Correlation | .028 | .173 |
|  | Sig. (2-tailed) | .863 | .285 |
| SASB Cl 5 | Pearson Correlation | -.163 | -.262 |
|  | Sig. (2-tailed) | .316 | .103 |
| SASB Cl 6 | Pearson Correlation | .107 | .067 |
|  | Sig. (2-tailed) | .512 | .680 |
| SASB Cl 7 | Pearson Correlation | .387* | .637** |
|  | Sig. (2-tailed) | .014 | .000 |
| SASB Cl 8 | Pearson Correlation | .576** | .722** |
|  | Sig. (2-tailed) | .000 | .000 |

Depression (BDI) and SASB Cls (Intrapsychic behaviours).

general well-being of the caregiver. This may suggest that if a high level of social support becomes available to everyone to lower CBI, it will benefit their overall health in the long run. The level of unmet supportive care needs of caregivers is highly associated with intrapsychic behaviours of structure of personality.

In fact interpersonal relationships are determined by the individual's experience and personality. So the caregivers with intrapsychic problematic traits and, consequently, in interpersonal relationships, should undergo closer surveillance. The ability to deal with relationships, even problematic ones, is seen as a resource that could reduce vulnerability to distress or buffer the adverse psychological effects of care burden duration and family history. Many studies have shown that the amount of social support available in the environment for individuals reduces their chances for developing any negative outcome in their health and mood. So the caregivers with SASB problematic intrapsychic traits should be regarded as having a high risk of worse social support and depression and could be followed up and screened. Lifestyle and interpersonal style and the existing social support of the individual caregiver (subjective and objective loneliness and or / isolation) must also be taken into account in planning multidimensional interventions. Assessing personality traits before providing an education program is highly recommended for caregivers. This assessment could improve the quality of personalized education programs and better meet caregiver needs.

Personality trait was found to be an important factor correlated with depression and care burden, suggesting the following considerations:

The problematic and depressed caregivers should undergo closer surveillance than caregivers without these intrapsychic problematic behaviours. Early identification of these caregivers could be necessary in order to provide mental health professionals with the opportunity of proactive intervention [18, 19].

The presence of the association between CBI and depression in family caregivers and problematic intrapsychic modalities may suggest targeted interventions aimed at helping a positive adaptation of caregiving difficulties.

This intervention could help to overcome depression and the maladaptive intrapsychic modalities, promoting self-care even in a situation such as caring for a demented patient [37]. It may also favor the adaptation to their condition of being burdened by improving their natural defenses [18, 19]. On the basis of our results we hypothesize that Behavioral Activation (BA) therapy and/or Holistic Psychotherapy with mindfulness might be a more suitable intervention [18–20, 38].

## Conclusions

The study suggests that screening for intrapsychic behaviors of the structure of personality linked to depression and the association of depression with CBI may help early detection of a non adaptation to the caregiver burden [1, 8].

Other studied are required to further explore new methods to support the specific needs of the patients, including the interpersonal ones.

This study has several limitations. The first limitation is the small sample. Moreover all caregivers were caring for patients affected by Alzheimer's disease, and care is needed when extrapolating these results to other cognitive impairment diseases. A sampling bias was present in the data because all the subjects attended only one institution and thus were not representative of caregivers in general. Our results provide a snapshot of depression, care burden and, intrapsychic and interpersonal behaviors. The results may differ during the different phases or at other points in the disease journey. The caregiver's experience may evolve naturally over time and we don't have enough data to make these comparisons. Further studies may highlight these issues.

## Supporting information

**S1 Appendix. SASB—Structural Analysis of Social Behaviours: Rules for determining intrapsychic behaviours.**
(DOC)

## Author Contributions

**Conceptualization:** Anna Vespa, Roberta Spatuzzi, Maria Velia Giulietti.

**Data curation:** Anna Vespa, Roberta Spatuzzi, Martina Penna, Maria Velia Giulietti.

**Investigation:** Martina Penna.

**Methodology:** Anna Vespa, Paolo Fabbietti.

**Project administration:** Anna Vespa.

**Software:** Martina Penna.

**Supervision:** Anna Vespa, Paolo Fabbietti, Maria Velia Giulietti.

**Writing – original draft:** Anna Vespa, Roberta Spatuzzi, Paolo Fabbietti, Maria Velia Giulietti.

**Writing – review & editing:** Anna Vespa.

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
