## [Decision Letter · Decision Letter 0]

15 Feb 2021

PONE-D-20-38970

Association between care burden, depression and personality traits in Alzheimer’s caregiver: a pilot study

PLOS ONE

Dear Dr. Vespa,

Thank you for submitting your manuscript to PLOS ONE. After careful consideration, we feel that it has merit but does not fully meet PLOS ONE’s publication criteria as it currently stands. Therefore, we invite you to submit a revised version of the manuscript that addresses all the points raised during the review process.

We look forward to receiving your revised manuscript.

Kind regards,

Gianluigi Forloni

Academic Editor

PLOS ONE

Journal Requirements:

3. In line with PLOS' guidelines regarding the description and reproducibility of methods (https://journals.plos.org/plosone/s/criteria-for-publication#loc-3), please provide further detail on how the questionnaires and testing procedures were administered.

- Please improve statistical reporting and ensure that numbers are properly formatted, i.e., with a decimal point instead of a comma. Our statistical reporting guidelines are available at https://journals.plos.org/plosone/s/submission-guidelines#loc-statistical-reporting.

- Please also note that PLOS ONE does not copy edit accepted manuscripts (https://journals.plos.org/plosone/s/criteria-for-publication#loc-5), and so the language in submitted articles must be presented in an intelligible fashion and written in standard English. To that effect, please ensure that your submission is free of typos, grammatical errors, and is written entirely in English.

6. Thank you for submitting the above manuscript to PLOS ONE. During our internal evaluation of the manuscript, we found significant text overlap between your submission and the following previously published works:

- https://onlinelibrary.wiley.com/doi/abs/10.1002/gps.4232 ("Caregiver burden characterization in patients with Alzheimer's disease or vascular dementia", 2014, by D'Onofrio et al.)

Please revise the manuscript to rephrase the duplicated text, cite your sources, and provide details as to how the current manuscript advances on previous work. Please note that further consideration is dependent on the submission of a manuscript that addresses these concerns about the overlap in text with published work.

Reviewers' comments:

Reviewer's Responses to Questions

**Comments to the Author**

1. Is the manuscript technically sound, and do the data support the conclusions?

Reviewer #1: Yes

Reviewer #2: No

2. Has the statistical analysis been performed appropriately and rigorously? 

Reviewer #1: I Don't Know

Reviewer #2: No

3. Have the authors made all data underlying the findings in their manuscript fully available?

Reviewer #1: Yes

Reviewer #2: Yes

4. Is the manuscript presented in an intelligible fashion and written in standard English?

Reviewer #1: No

Reviewer #2: No

5. Review Comments to the Author

Reviewer #1: This descriptive survey design study investigates the relationship of caregiver burden, to the eight clusters of the SASB, depression, and everyday cognition, for caregivers of family members with Alzheimer's disease and the mini-mental examination of the care receiver. The sample consisted of 59 caregivers caring for a family member diagnosed with Alzheimer's disease. Psychologists collected data. Eighty-eight caregivers were approached, and seventy-three agreed to participate. In particular, the researchers tried to identify psychological vulnerabilities (intrapsychic behaviors) and investigate the role care burden has with depression and specific conditions (neuropsychiatry symptoms). They noted that this study is innovative in that it is the first to look at the relationship between intrapsychic behavior with caregiver burden. The results of this study could lead to tailored caregivers’ interventions.

It is my understanding that the statistical analysis has been performed appropriately. The sample size was small, and the number of variables and subscales used in the analysis extensive. I don't think there was any other analysis but correlations appropriate for this study. Still, additional analysis may be done that I am unaware of.

The research team found that CBI correlates with the total NPI, specifically delirium, depression, anxiety apathy, aberrant motor activity, sleep disorders, appetite, and nutrition disorders. This information has been presented in past research studies studying Alzheimer's disease caregivers.

CBI is related to depression and intrapsychic and interpersonal behaviors of caregivers. This descriptive survey can not specify the relationship's direction, so some caregivers may enter the caregiving experience depressed, thus influencing caregiver burden. Nevertheless, there were some interesting findings. One weakness of the researchers' introduction of the SASB model is that they do not explain the model or the three underlying dimensions and how the assessment of individuals (self or observers) work. Because of the nature of the design, the statement that intrapsychic behaviors pre-exist the caregiver needs requires further exploration. I wonder if studies that support this hypothesis's probability could be cited. The following points if addressed could strengthen the manuscript.

• What were the reasons from the caregivers given for not participating?

• What was the training of the psychologists on the SASB assessment surveys?

o Were the psychologists part of the research team or employees of the hospital?

o How were they recruited?

• An explanation of the SASB Module and its three underlying dimensions is important to include.

• There are sections in the manuscript that needs significant editing. Some phrases are a bit confusing but

understandable. The authors need to look at the document again.

• Some areas need major editing; see the following.

o We believe that the Person-centered approach, with clusters analysis, conceptualizes the personality as "a related

system of different traits" of personality allowing the identification of latent classes (subgroups) of individuals with

distinct personality size profiles, is the more appropriate in this context [18, 19] as it allows us to evaluate

problematic intrapsychic experiences (page 8) what does this mean?

o Interpersonal reaction-interpretation may be adequate or inadequate. Clarify meaning

o The paper is well documented, but on page 8, On page 8, six-lines down, there are several statements related to

caregivers' restriction of activities with no citations. They have them in the reference section. They need to include

some in this area.

Thank you for the opportunity to review this manuscript.

Reviewer #2: This paper addresses an important topic ie the relationship of personality factors to caregiver burden. The measurement tools are generally appropriate however there are a number of concerns which limit the reliability of the findings. Most important the analysis describes a large number of correlational outcomes with many variables, the reliability and validity of which cannot be supported by the very small sample size of each item. Variables are not independent but there is no correction attempted. No sample size justification is offered. The discussion/conclusions focus only on the correlations between the global outcomes for key measures which appear to be based on the whole sample rather than subsets as as described in the tables. The authors have not clarified this discrepancy in how the data is presented.

It is known that living with the person with dementia is a determinant of caregiver burden. This data is lacking.

The data in the tables is hard to interpret as there is not provided for each outcome. I note that the percent of each gender of caregiver does not add to 100%. I do not know if this is the only problem with the data as presented in the tables.

Conclusions based on the correlational analysis eg relationship differences (e.g. spouses (15% = N of 9 versus non spouses N=50) end up relying on likely insufficient numbers for a reliable analysis and conclusions especially since the sensitivity of the CBI measure is not clarified in the paper.

It appears that the paper was translated from Italian. There is one sentence still left in the original. If the paper is published I recommend that it be carefully edited for English language usage and clarity.

I suggest that the paper be rewritten after sample size justifications have been determined and focus only on the data which can be statistically supported by the method. This may still yield valuable outcomes and conclusions about the relationship between personality and burden.

6. PLOS authors have the option to publish the peer review history of their article (what does this mean?). If published, this will include your full peer review and any attached files.

Reviewer #1: No

Reviewer #2: No

---

## [Author Response · Author response to Decision Letter 0]

22 Mar 2021

Dear Reviewers 

I hope I have fully answered the requests of both the Reviewers. 

Thank you for your comments.

I am especially grateful for the suggestions very useful for future investigations concerning the relationship between personality, care burden and quality of life of caregivers of patients with Alzheimer's dementia.

Best regards 

Anna Vespa 

(I have responded to every correction request from reviewers- see attach file)

---

## [Editor Report · Decision Letter 1]

4 May 2021

Association between care burden, depression and personality traits in Alzheimer’s caregiver: a pilot study

PONE-D-20-38970R1

Dear Dr. Vespa,

We’re pleased to inform you that your manuscript has been judged scientifically suitable for publication and will be formally accepted for publication once it meets all outstanding technical requirements.

Kind regards,

Gianluigi Forloni

Academic Editor

PLOS ONE
---

## [Editor Report · Acceptance letter]

20 Sep 2021

PONE-D-20-38970R1 

Association between care burden, depression  and personality traits in Alzheimer’s caregiver: a pilot study 

Dear Dr. Vespa:

I'm pleased to inform you that your manuscript has been deemed suitable for publication in PLOS ONE. Congratulations! Your manuscript is now with our production department. 

Kind regards, 

on behalf of

Dr. Gianluigi Forloni 

%CORR_ED_EDITOR_ROLE%

PLOS ONE